# Circulating Pro- and Anti-Inflammatory Metabolites and Its Potential Role in Rheumatoid Arthritis Pathogenesis

**DOI:** 10.3390/cells9040827

**Published:** 2020-03-30

**Authors:** Roxana Coras, Jessica D. Murillo-Saich, Monica Guma

**Affiliations:** 1Department of Medicine, School of Medicine, University of California, San Diego, 9500 Gilman Drive, San Diego, CA 92093, USA; rcoras@health.ucsd.edu (R.C.); jdmurillosaich@health.ucsd.edu (J.D.M.-S.); 2Department of Medicine, Autonomous University of Barcelona, Plaça Cívica, 08193 Bellaterra, Barcelona, Spain

**Keywords:** metabolomics, microbiome, diet, lifestyle, circulating

## Abstract

Rheumatoid arthritis (RA) is a chronic systemic autoimmune disease that affects synovial joints, leading to inflammation, joint destruction, loss of function, and disability. Although recent pharmaceutical advances have improved the treatment of RA, patients often inquire about dietary interventions to improve RA symptoms, as they perceive pain and/or swelling after the consumption or avoidance of certain foods. There is evidence that some foods have pro- or anti-inflammatory effects mediated by diet-related metabolites. In addition, recent literature has shown a link between diet-related metabolites and microbiome changes, since the gut microbiome is involved in the metabolism of some dietary ingredients. But diet and the gut microbiome are not the only factors linked to circulating pro- and anti-inflammatory metabolites. Other factors including smoking, associated comorbidities, and therapeutic drugs might also modify the circulating metabolomic profile and play a role in RA pathogenesis. This article summarizes what is known about circulating pro- and anti-inflammatory metabolites in RA. It also emphasizes factors that might be involved in their circulating concentrations and diet-related metabolites with a beneficial effect in RA.

## 1. Introduction

Rheumatoid arthritis (RA) is an autoimmune inflammatory arthritis that affects approximately 1% of the world’s population. It is a potentially debilitating disease that affects women two to three times more frequently than men [1]. It is characterized by pain and swelling in joints and produces irreversible joint damage that negatively affects patients’ quality of life in the absence of treatment. In our clinical practice, patients often mention changes in their symptoms after the consumption or avoidance of certain foods and inquire about the adequate type of diet for this disease. However, there is very little knowledge on how diet or specific ingredients affect pain and inflammation in RA. Recently, a lot of research on diet, gut microbiome, and gut-microbe-derived metabolites has focused on explaining how this diet–microbiome-metabolomic axis can explain different symptoms and overall health status.

Several studies have employed different analytical methods (mass spectrometry, MS, nuclear magnetic resonance, NMR) to characterize the metabolomic profile in the blood (serum or plasma), urine, or synovial fluid in patients with rheumatoid arthritis. Due to the heterogeneity of the methods that were used, the results of most of the studies are not comparable; however, there are metabolites with similar changes across multiple studies (Figure 1 and Table 1). The objective of this work is to review the existing evidence for the relationship between diet, metabolites, and inflammation in RA.

## 2. Factors That Influence Circulating Metabolites and Their Potential Role in Rheumatoid Arthritis

Metabolites reflect an organism’s state, which results from the interaction of internal and external factors, such as genetic and environmental/lifestyle factors, respectively. In disease states, the circulating metabolites are also affected by the pathological processes, and there are already well-studied metabolites that are considered to be disease reporters, like the increase of blood glucose levels in diabetes mellitus. In systemic diseases such as RA, the abnormal circulating metabolomic profile might reflect genetic predisposition, local inflammation, comorbidities, and several environmental factors including diet, smoking, or microbiome (Figure 2).

### 2.1. Diet

Amongst environmental factors, diet is one that directly affects circulating metabolites. For example, essential amino acids (histidine, isoleucine, leucine, lysine, methionine, phenylalanine, threonine, tryptophan, and valine) and essential fatty acids (alpha-linolenic acid and linoleic acid) come from the diet. Of interest, some of these essential nutrients were found to be low in RA patients, including linoleic acid, and several amino acids (Table 1), suggesting a link between diet and inflammation in RA.

Epidemiological studies have shown a relationship between diet and RA; thus, some of the metabolomic changes observed in several fluids (serum/plasma or urine) in early arthritis could be related to differences in dietary patterns between RA patients and the healthy population. A study that included a large number of patients (15770 adult males and females) found that patients with arthritis (including RA and osteoarthritis) had lower quality diets compared to people without arthritis, based on HEI-2015, a healthy eating index created by the USDA and based on the Dietary Guidelines for Americans [9]. Patients with arthritis consumed less fruit, vegetables, greens and beans, whole grains, seafood, and plant protein, but more added sugars, saturated fats, and empty calories compared to those without arthritis [10]. The association of poor dietary quality with RA was also observed in other studies, in which RA patients had an inadequate intake of fruit, vegetables, dairy, fatty acids, and whole grains [11,12,13]. A study in a Chinese population found that RA patients were consuming different amounts of chicken, fish, mushrooms, beans, citrus, dairy products, and organ meats than healthy controls [14]. Another study that included a white population found that both women and men on a nonvegetarian diet were at higher risk of developing RA [15]. Hu et al. analyzed the cohort of women included in the Nurses’ Health Study and Nurses’s Health Study II that were followed from 1984 to the present-day, and found that good dietary quality, moderate alcohol consumption, and low intake of red meat were associated with a lower rate of RA incidence [13]. 

In the past, due to advances in the field of metabolomics, efforts have been made to predict food intake by measuring blood/urine/fecal metabolites. Two main techniques are being used: MS coupled with liquid- or gas-phase chromatography and proton (^1^H) NMR [16]. There are numerous metabolomics studies that have identified candidate biomarkers for different dietary patterns, as well as for different types of foods, ranging from meat to fruits and vegetables. Table 2 shows a summary of foods and the metabolites that have been found to be markers of their intake using metabolomics. Several studies have also found metabolites related to dietary patterns, like Mediterranean, high fat, or Western diets [17,18,19,20,21].

The identification of food biomarkers is an ongoing process; a consensus has not been reached as to which metabolites would be the most adequate biomarkers for different types of foods. Moreover, some metabolites are markers of categories of food, not being able to discriminate between the exact types of foods being analyzed (1-methylhistidine and 3-methylhistidine are found in meat and are not useful in discriminating between types of meat). It is possible that for some foods, a combination of metabolites would be more suited as a marker than a single metabolite. Unfortunately, as of now, the metabolomic studies in RA (Table 1) have not collected food intake data nor used the same metabolomic platforms, making it difficult to associate specific food intake with metabolic changes in RA patients. However, some metabolites from Table 1 and Table 2 suggest an interaction between circulating metabolites and diet. For instance, some of these studies [5,59] showed higher levels of carnitine and taurine in RA patients, which are potential biomarkers of meat intake.

### 2.2. Drugs

Researchers have used a metabolomics approach to evaluate the changes in circulating metabolites from drugs used in RA treatment (Table 3). The study of these changes might help to understand RA pathogenesis, since the therapeutic effects of these drugs could potentially be driven by metabolic changes either by normalizing their abnormal values or by increasing anti-inflammatory metabolites. For instance, using a targeted metabolomic approach, Fu et al. compared the effect of oral glucocorticoids (GC) on serum polar lipids and observed an increase in lysophosphatidylcholines (LPC) and lysophosphatidylethanolamines (LPE) in females but not in male patients with RA [7]. GC inhibits phospholipase A, a key enzyme that hydrolyzes membrane phospholipids which is increased in inflammatory tissues. The effect of GC on phospholipase A will likely modify the phospholipid profile. Of interest, polyunsaturated acyl LPC and LPE presented an anti-inflammatory effect on animal models [60]. The effect of low dose GC (<10 mg/day) on arginine metabolism and cardiovascular risk in RA patients was also studied [61]. This study from Australia that included 36 RA patients, 18 of which were on GC (GC users) and 18 that were not receiving GC (non-GC-users), found that asymmetric dimethyl arginine (ADMA) and symmetric dimethyl arginine (SDMA) levels were lower in patients on chronic GC compared to non-GC users, suggesting that long-term treatment with GC had an improved endothelial function and a cardiovascular protective effect. by modulating arginine metabolism [61]. 

Wang et al. [59] studied the change of the plasma metabolic profile in 29 RA patients after the initiation of treatment with methotrexate (14 patients) or a combination of methotrexate with a Chinese medicinal herb (15 patients). They found decreased levels of several amino acids (tryptophan, threonine, histidine, methionine, and glycine) as well as other metabolites (carnitine, hypoxanthine, cytosine, uracil, and uric acid), while taurine, aspartate, alanine, lactic acid, adenosine, and guanine were significantly increased in RA patients compared to controls. Interestingly, the treatment with methotrexate brought the levels of all these metabolites back to normal levels, suggesting a causative role of these amino acids in RA pathogenesis. The combination of MTX with tripterygium glycosides tablets was more effective in obtaining these results compared to monotherapy with MTX. Although more data is needed to link amino acid changes to abnormal immune response in RA, data in immune cells suggest a direct link between amino acid metabolism and T cell and macrophage responses by promoting and modulating inflammation, which could potentially be involved in RA pathogenesis [62,63,64,65]. In RA, tryptophan is the substrate of indoleamine-2,3-dioxygenase IDO2, which was demonstrated to be required for the activation of CD4+ Th cells, the production of pathogenic autoantibodies, and the subsequent development of arthritis in a KRN mouse model of arthritis [66,67,68]. This offers a possible explanation for the decrease in tryptophan levels that is then reversed by the addition of methotrexate. On the other hand, levels of S-adenosy-L-homocysteine, 5-formyltetrahydrofolate, and 5-methyltetrahydrofolate were similar between controls and RA patients before treatment, and decreased after 3 months of methotrexate, pointing to these methotrexate-associated metabolites as adherence biomarkers [59]. 

TNF is a potent pro-inflammatory cytokine that plays key role in cell metabolism, including glucose and lipid metabolism [72]; thus, changes in metabolic profile are expected after the administration of a TNF inhibitor. The first study that evaluated the changes in the metabolic profile of 16 RA and psoriatic arthritis (PsA) patients after TNFi treatment (etanercept and infliximab) used urine samples. The study described increases in hippuric acid, citrate, and lactic acid after infliximab treatment, while increases in choline, phenylacetic acid, urea, creatine, and methylamine were seen after etanercept treatment [71]. Another group evaluated the serum metabolomic profile in 20 patients with RA before and after treatment with TNFi (etanercept or adalimumab). Of the 20 patients, 55% of patients had a moderate EULAR response, while only 20% reached a good response. At 3 months posttreatment, 3-hydroxyisobutyrate, lysine, acetoacetate, acetylphosphocholine, creatine sn-glycero-3-phosphocholine, histidine, and phenylalanine levels decreased, while leucine, acetate, betaine, and formate levels increased, but they did not reach those of the healthy control [69]. The changes of the serum metabolic profile in response to treatment with a TNFi, etanercept, in 27 patients with active RA were also evaluated by Priori et al. These patients were receiving concomitant therapy with GC and disease-modifying antirheumatic drugs. After 3 months of treatment, isoleucine, leucine, valine, alanine, glutamine, tyrosine, and glucose levels were found to be increased in good responders as defined by EULAR-ESR criteria, whereas 3-hydroxybutyrate levels were reduced [70]. The decrease of 3-hydroxybutirate, acetoacetate, and acetylphosphocholine levels suggests a modulation of lipid metabolism after TNF inhibition, especially in responders. In addition, the increase of glucose and other amino acids suggests a decrease of glucose and amino acid metabolism by the inflamed tissues.

### 2.3. Comorbidities

RA patients present several comorbidities including obesity, metabolic syndrome, and sarcopenia, probably triggered by a disbalance of proinflammatory cytokines including TNF and IL-6 among other causes [73,74,75,76,77], that will modify the circulating metabolites [78]. Several studies have investigated circulating metabolic changes related to the metabolic syndrome and obesity [79,80]. Of interest, a lot of circulating metabolites that are different in RA patients compared to controls could be related to associated metabolic syndrome, since choline metabolism (especially TMAO and carnitine), aminoacids (alanine, glutamine, glutamate, arginine, aspartate, asparagine, histidine, methionine, cysteine, lysine, branched-chain amino acids (BCAA), phenylaniline, tyrosine, and tryptophan) and phospholipids (phosphatydilcholines) also change in those with metabolic syndrome [80]. Several works on muscle mass have also suggested that some circulating metabolites can be biomarkers of muscle mass and sarcopenia [81]. Even though both fat tissue and muscle, as well as associated immune cells in these inflamed tissues, can be sources of metabolites, it is unknown how much they can contribute to the pool of circulating metabolites. For example, studies measuring the metabolomics profile in visceral adipose tissue and serum from obese patients found low correlations between serum and adipose tissue metabolites [82]. On the other hand, we can speculate that there might be a competition between inflamed tissues (adipose tissue vs. synovial tissue) for the uptake of circulating anti-inflammatory metabolites.

### 2.4. Sex and Age

Several epidemiological studies have shown differences in metabolite concentrations according to sex and gender. A cross-sectional study in urine samples showed that some metabolites from the tricarboxylic acid cycle (TCA) cycle such as citrate and fumarate were elevated in women, while carnitine, acetylcarnitine, acetone, and creatinine were higher in men [83]. In addition, Fan et al. found that 2-hydroxyglutaric acid, α-ketoglutarate, and 2-oxyglutaric acid were higher in women. However, UDP-glucoronic acid was higher in men, suggesting that this could be linked to sex hormones [84]. Another study showed differences between sex and metabolic profile in serum, suggesting that glycine, serine, and sphingomyelines are upregulated in women, and ornithine, arginine, acyl carnitines, and amino acids derived from glutamine pathway are elevated in males [85]. Finally, a longitudinal cohort of adults showed a positive correlation of levels of glutamine, tyrosine, long chain fatty acids, acyl-carnitines, and sphingolipids, and a negative correlation of histidine, tryptophan, threonine, serine, and leucine levels with age [86].

### 2.5. Smoking and Exercise

Smoking is a known risk factor for RA and is associated with an increased risk of more severe arthritis, and less likelihood of achieving remission. Smoking also decreases the effectiveness of some disease-modifying antirheumatic drugs (DMARDs) [87,88]. The exact reason of these associations is not well understood, although the effect of smoking on immune cells, and cytokine production, and the increase of oxidative stress that it causes, have been described [89,90,91], and these likely affect the immune response in RA. Metabolomics has also identified blood biomarkers associated with chronic tobacco smoking. One study performed on a large number of healthy participants (892) from around the world found an association between smoking and three well-established nicotine metabolites (cotinine, hydroxycotinine, and cotinine N-oxide), and an additional 12 xenobiotic metabolites involved in benzoatic (e.g., 3-ethylphenylsulphate) or xanthine metabolism (e.g., 1-methylurate), three amino acids (o-cresol sulphate, serotonin, indolepropionate), two lipids (scyllo-inositol, pregnenolone sulphate), four vitamins or cofactors, and one carbohydrate (oxalate) [92]. Several of these metabolites, especially nicotine-derived metabolites, have been described to modulate the immune response [90], and other metabolic changes could be involved in smoking-induced methylation changes in immune cells [93]. Another study looked at the immediate effects of smoking on the metabolic profile. Thirty-one metabolites were shown to be acutely affected by cigarette smoking, including menthol-glucuronide, the reduction of glutamate, oleamide, and 13 glycerophospholipids. Moreover, detailed analysis revealed changes in 12 cancer-related metabolites, notably related with cAMP inhibition [94]. Since a known mechanism of methotrexate in treatment of RA is to induce an increase of cellular cAMP [95], the inhibition of this metabolite by smoking could explain the decrease in the effectiveness of this drug in RA.

Exercise is another factor that might change the metabolomic profile. However, these changes depend of the quantity and type of exercise. For example, in people who exercise more than 2 h per day, some metabolites, including medium and long fatty acids, ketones, sulfated bile acids, palmitate, linoleate, stearate, and palmitoleate, increased two-fold in their plasma concentration. Decreases of pyruvate and lactate, among others intermediates of TCA, have been reported after a short running period [96]. The reader can find an extensive review of these changes in a recently published review [96] about metabolic changes after exercising. 

### 2.6. Genetics: Polymorphisms and Metabolism

Genome-wide association studies (GWAS) uncovered multiple loci that are associated with the level of metabolites, which involve a large number of metabolic pathways, indicating widespread genetic influences on the human metabolome (Figure 3). The loci that have been described involve amino acids, intermediates of lipid metabolism, including sterols, carnitines, and intermediates of inositol and fatty acid metabolism, intermediates of purine and pyrimidine metabolism, glucose homeostasis, and vitamin and cofactor levels [97,98,99,100]. Polymorphisms in these metabolite-associated genes were also described in RA GWAS. In Figure 3, we put together a summary of metabolism-related genes described in genome-wide association studies (GWAS; https://www.ebi.ac.uk/gwas/). Polymorphisms in the genes underlined in red were found to be associated with RA. These genes are mostly related to lipid metabolism. Interestingly, lipid metabolites are considered pro-inflammatory metabolites (see Section 3), and higher levels of lipids were described in serum of RA patients compared to control subjects (Table 1). *DLG2* (Discs Large MAGUK Scaffold Protein 2), which was found to be associated with glycerophospholipid metabolism [101], was also found to be related to response to TNF inhibitors in RA patients [102]. *FADS1 and 2* (Fatty Acid Desaturase) and *BLK* (BLK Proto-Oncogene, Src Family Tyrosine Kinase), involved in fatty acid metabolism, and *STAG1* (Stromal Antigen 1) and *FCGR2B* (Fc Fragment Of IgG Receptor IIb), involved in lipoprotein metabolism, were found to be associated with susceptibility to developing RA in several studies [103,104,105,106,107]. SLC22A4, a transporter related to isovaleryl/carnitine, was found to be associated with RA in a Japanese population [108], but not in a Canadian one [109]. Finally, Geiger et al. described 2 SNPs (single nucleotide polymorphism), rs9309413 and rs4775041, found on PLEK (Pleckstrin) and LIPC (Hepatic Triacylglycerol Lipase) genes, associated with sphingomyelin associated and phosphatidylethanolamine (PE) [110], that were associated with RA in a previous study [111]. Little is known about the role of these genes in inflammation and autoimmunity, so more studies are needed to determine whether some of these pathways are critical for the pathogenesis of RA.

### 2.7. Gut Microbiome/Absorption

The gut microbiome represents the collection of microbes that inhabit the intestines. Its composition is shaped by several factors, like genetics, age, delivery pattern, diet, antibiotic use, and other treatments [112,113,114,115]. It can also be modulated by prebiotics [116,117], probiotics [118], and fecal microbiota transplantation. Bacteria in the gut are important not only in the absorption of certain vitamins and in the synthesis of bile acids, but they also have the potential to modify circulating pro- or anti-inflammatory mediators, since they are involved in the metabolism of some dietary components [119]. For example, trimethylamine-*N*-oxide, a pro-inflammatory metabolite that derives from choline and carnitine present in red meat, eggs, and dairy products, is produced by *Prevotella copri* among other bacteria [120,121]. An increased abundance of *Prevotella copri* was found in new-onset untreated RA patients, suggesting *P. copri* may be pathogenic in this disease [122]. In contrast, bacteria that have an almost exclusive saccharolytic metabolism, such as lactobacilli and bifidobacterial, are considered potentially beneficial [123], since they produce a variety of tryptophan catabolites (indole, tryptamine, indoleethanol (IE), indolepropionic acid (IPA), indolelactic acid (ILA), indoleacetic acid (IAA), skatole, indolealdehyde (IAld), and indoleacrylic acid (IA)) which are critical for intestinal homeostasis by decreasing intestinal permeability [124]. In addition, some of these catabolites enter the bloodstream and may have anti-inflammatory and anti-oxidative effects [124]. The microbial degradation of whole-grain complex carbohydrates increases short-chain fatty acids (SCFA; butyrate, acetate and propionate), which were also shown to be beneficial to the intestinal immune response [125]. Microbial bile acid metabolites have recently been linked to colonic homeostasis [126].

The modulation of the microbiome through diet interventions is a potential strategy in the treatment of diseases, since microbiome alterations are related to disease, i.e., inflammatory bowel disease, obesity, cardiovascular diseases, autoimmune diseases, and others. It seems that the microbiome response to diet is variable and is highly influenced by the subject’s baseline microbiome. Several studies found differences in the baseline microbiome of responders versus nonresponders to different diet interventions. Additionally, individuals with differing bacterial gene richness appear to have differing baseline gut microbiota communities that respond distinctively to a given dietary intervention which will influence the diversity of the gut microbe-derived specialized metabolites and circulating metabolites [127,128,129,130,131,132,133,134,135,136]. 

### 2.8. Metabolite Released from or Uptaken by Inflamed Tissues

Another potential factor that determines the concentrations of the circulating metabolites is represented by the release of metabolites from the inflamed joint or their uptake by the synovium. Little is known about metabolic or lipidomic profiling of synovial tissue [137,138]. In addition, no study has, to date, evaluated the relation between circulating metabolites in serum or plasma and synovial metabolites, although there might be a correlation. For instance, the synovial tissue of RA patients presents an enhanced level of lactate compared to noninflamed synovial tissue [138], which suggests an increase in the anaerobic cellular metabolism of resident cells [139,140]. Lactate has also been one of the metabolites described to be upregulated in patients with RA [5]. Of note, inflammatory pathways increase the expression of nutrient transporters [141,142,143,144]; therefore, this highly metabolic tissue will consume high amounts of metabolites, either to feed the increased metabolism of activated cells (pro-inflammatory metabolites) or to resolve inflammation (anti-inflammatory metabolites); this could be reflected by a decrease of circulating metabolites described in RA (Table 1 and Figure 1): such as glucose and amino acids (alanine, serine, methionine, threonine, leucine, valine, isoleucine, aspartate, phenylalanine, tyrosine, and proline) [6,145]. 

Fibroblast-like synoviocites (FLS), key cells in the pathogenesis and progression of RA, have an activated metabolism and can potentially release metabolites into the bloodstream [146]. Ahn et al. [147] characterized the intracellular metabolic profile of RA and osteoarthritis (OA) by an untargeted metabolomic approach using GC/TOF-MS. The results revealed that a high number of metabolites were increased in RA compared to OA FLS; these metabolites were amines (inosine, urate, 5′-deoxy-5′-methylthioadenosine, guanine, benzamide), fatty acids (behenic acid, palmitoleic acid, arachidic acid, oleic acid, myristic acid, stearic acid, palmitic acid, octadecanol, linoleic acid, lauric acid), phosphates (glucose-6-phosphate, phosphogluconic acid, adenosine-5-monophosphate, phosphate, fructose-6-phosphate), organic acids (aspartate, adipate, 2-ketoisocaproate 3-phenyllactate, 2-hydroxyvaleric acid, phenylacetate, glycolate, oxalate, benzoate), amino acids (asparagine, glutamine), sugars and sugar alcohols (lactose fucose, mannose) and salicylaldehyde. Other metabolites, mostly amino acids (isoleucine, leucine, histidine, valine, ornithine, lysine, methionine sulfoxide, tryptophan, N-methylalanine, tyrosine, phenylalanine, citrulline, oxoproline, threonine, serine) were decreased in RA compared to OA FLS. At the same time, the glycolysis and pentose phosphate pathways were more activated in RA than OA FLS. RA FLS are aggressive cells, similar to cancer cells, and require high amounts of energy to fulfill their pathogenetic functions in RA, which include proliferation, migration, and invasion [148,149]. 

Macrophages and T cells are the other dominant type of synovial cells in the inflamed joint, and are important in the progression of the disease, with their abundance being correlated with disease activity but also response to treatment [150]. Similar to the FLS, activated macrophages and T cells also rely on glycolysis and have alterations of the TCA cycle [150], which is consistent with the high levels of lactic acid, citrate, and succinate found in the synovial fluid of RA patients [151]. Although metabolic profiling of RA synovial macrophages and T cells hasn’t yet been undertaken, they are probably a source of circulating metabolites, while metabolites also exert their effect on synovial cells [152,153,154,155]. 

## 3. Evidence for a Pro-/Anti-Inflammatory Role of Metabolites in RA

RA is a chronic autoimmune disease, with a systemic immune response to autoantigens that may exist years before the onset of clinical symptoms, and a local immune activation of the synovial tissue which becomes inflamed, hyperplastic, and invasive of local cartilage and bone [156]. Whether or not pro- or anti-inflammatory metabolites play a role in RA pathogenesis is still unknown. However, the explosive growth of the field of tissue immunometabolism and its description of multiple critical metabolic pathways in the activation and differentiation of immune cells such as T and B lymphocytes, macrophages, dendritic cells, and fibroblasts, among others (see reviews in [157,158,159,160,161]), suggests that most of the metabolites involved in the immune response can also be important in RA. Here, we describe pro- and anti-inflammatory metabolites associated with RA pathogenesis (Figure 4). 

### 3.1. Pro-Inflammatory Metabolites

Choline and Trimethylamine-N-oxide (TMAO). Metabolites related to the choline pathway were identified in several studies in synovial tissue, synovial fluid, and blood (serum/plasma) samples in both animal models and human studies. Diet is the main source of choline [162], whose metabolites (trimethylamine-N-oxide, TMAO) have already been related with cardiovascular inflammation [121,163]. Choline and other dietary trimethylamine (TMA) containing species like carnitine are metabolized to TMA by the gut microbiota. TMA is subsequently oxidized by at least one member of the flavin-containing monooxygenases, FMO3, forming trimethylamine-N oxide (TMAO), which is then released into circulation [164]. TMAO is a candidate biomarker for meat and fish intake, as can be seen in Table 2. 

Despite being so well studied in relation to cardiovascular inflammation, we were not able to find studies evaluating the role of TMAO in RA. Our group found that serum TMAO was associated with measures of joint (tender joint count, swollen joint count, DAS28-CRP) and skin inflammation (body surface area affected by psoriasis) in a small cohort of patients with psoriasis and PsA [165]. The increased TMAO in patients with psoriasis and PsA, two diseases associated with metabolic syndrome [166], could be due either to an increased activity of FMO3, which has been described to be upregulated in obesity [167], but also to changes in the microbiome composition, which is an intermediate component of TMAO synthesis. TMAO, as well as choline, was found to be increased in serum samples in the murine K/BxN model of arthritis compared to control mice [168]. Choline is also a nutrient uptaken by the cells and metabolized via the Kennedy pathway, during which several phospholipids that function as signaling molecules are produced, such as glycerol-phosphocholine (GPC), phosphocholine, phosphatidylcholine (PC), lyso-PC, diacylglycerol, and lysophosphatidic acid [169]. Importantly, choline metabolism has been related to the RA FLS phenotype [170] and IL-1β secretion in macrophages [142]. 

BCAA (Branched-chain amino acids). Decreased levels of valine, leucine, and isoleucine were found in RA patients. Decreased levels of BCAA could be explained by low dietary consumption or by a higher intake of these amino acids by inflamed tissue. These are essential amino acids, so their source is the diet, and lately, they have been related to inflammation by inducing oxidative stress (via NADPH and Akt-mTOR signaling) and promoting the secretion of proinflammatory cytokines (IL-6, TNF) as well as the migration of peripheral blood mononuclear cells [171]. Branched chain aminotransferases1 (BCAT1), an enzyme that initiates BCAA metabolism, is the predominant isoform in human primary macrophages. Its action on leucine produces acetyl-CoA and glutamate, which enter the TCA cycle. Treatment of LPS and TNF stimulated human macrophages with ERG240, a leucine analogue that blocks BCAT1 activity, decreased oxygen consumption and glycolysis. Moreover, oral administration of ERG240 reduced the severity of collagen-induced arthritis in mice [172]. 

Glutamine is an amino acid used as a source to fuel metabolism. Glutaminase 1, the enzyme responsible of glutaminolysis, is upregulated in RA synovial fibroblasts [173], and inhibition of this enzyme decreased the aggressive phenotype of the FLS and improved the severity of arthritis in the SKG murine model of arthritis. 

Glycolytic intermediates: The RA joint is characterized by a shift of the aerobic oxidative phosphorylation to a glycolytic state, in which less ATP is produced but at a faster rate, to be able to ensure the necessary energetic requirements for the highly active cells. Metabolites related to the glycolytic pathway have been detected in several studies on animal models, as well as human metabolomics studies. 

Lactate is the end product of glycolysis, a metabolic pathway that is upregulated in activated FLS and macrophages. High concentrations of lactic acid are found in both blood and synovial fluid from inflamed joints in RA patients. Several studies have shown that lactate promotes the aggressive phenotype of FLS [174], the pro-inflammatory properties of macrophages [152,153,154], stimulates IL-17 secretion by CD4+ T cells and, at the same time, decrease CD4+T migration, which is related to the maintenance of a chronic inflammatory infiltrate [155,175]. Moreover, recently, a new lactate induced histone modification was described, lactylation, which correlates with the levels of lactate and is different from acetylation [176]. These findings require further study to evaluate their role in disease states, since altered epigenetic marks have been recently described in RA FLS [177].

Succinate is elevated in the synovial fluid of patients with RA [151]. The TCA metabolite promotes inflammation by stimulating IL-1β secretion in murine macrophages through HIF-1α [152]. Moreover, succinate activates NLRP3 inflammasome inducing IL-1β secretion by synovial fibroblasts in a rat model of RA [178]. It seems that succinate also plays a role in innate and adaptive immune responses. Researchers found that the genetic deficiency of Sucnr1, a succinate receptor expressed by immune cells, decreases trafficking of dendritic cells and reduces expansion of Th17 cells in the lymph nodes, reducing the symptoms of arthritis in the mouse antigen-induced arthritis model [179].

Itaconate, a macrophage activation marker, is thought to play an anti-inflammatory role, since it inhibits the succinate dehydrogenase-mediated oxidation of succinate, and through this, exerts anti-inflammatory effects in activated macrophages, as shown in an in vivo model of ischemia-reperfusion injury [180]. However, in an animal model of RA, higher levels of itaconate were found to be associated with high disease activity [181]. 

Cholesterol comes from the diet and its levels are increased in RA patients [2,6]; this was found to be predictive of RA in women, but not men [182]. Lipid metabolism is altered in RA, but cholesterol metabolism in RA has not been specifically studied. Interestingly, cholesterol was recently found to be high in OA chondrocytes, due to an increased uptake, upregulation of cholesterol hydroxylases, and increased production of oxysterol metabolites [183]. 

Free fatty acids (FFA) can be either taken from the diet (essential FA, alpha-linolenic acid -an omega-3 FA- and linoleic acid -an omega-6 FA-) or synthesized in the organism. it was suggested that they were proinflammatory, since they contribute to low-level inflammation in obese patients. FFA levels were higher in the serum of RA patients than in control subjects, and correlated with disease activity [184]. Frommer et al. showed that FFA contribute to the pathogenesis and damage in RA, OA, and PsA, since stimulation of FLS with oleic, palmitic, and linoleic acid induced the secretion of proinflammatory cytokine IL-6, the chemokines IL-8 and MCP-1, as well as the matrix metalloproteinases pro-MMP1 and MMP3 [185]. Arachidonic acid (ARA) is the precursor of classically described prostaglandins (PGE2), which are known to be involved in inflammation in general, but also in arthritis [186,187]. 

### 3.2. Anti-Inflammatory Metabolites

Polyunsaturated Fatty Acids (PUFA) Related Metabolites. Docosahexaenoic acid, DHA, and eicosapentaenoic acid, EPA have anti-inflammatory properties, mainly because they compete with ARA for the action of the enzymes (cyclooxygenase-COX, lypooxigenase-LOX, cytochrome P450) which results in a decreased production of ARA derived proinflammatory oxylipins and an increased production of DHA and EPA derived anti-inflammatory oxylipins (Figure 5) [188,189]. Several studies have described improved outcomes in RA patients after dietary intervention [188,190,191]. Decreased levels of EPA and DHA were described in Spanish RA patients, and were associated with higher disease duration, positivity for rheumatoid factor, erosive disease and with a worse response to TNF inhibitors [192]. Gene variants of the enzymes involved in the PUFA metabolism can determine the metabolic and clinical response to dietary intake of PUFA. For example, 5-lipoxygenase (ALOX5) gene variants were found to influence response to fish oil supplementation, changing the oxylipin profile and, consequently, having a different effect on cardiovascular risk [193]. Another study checked the association between genetic variants of ALOX5, ALOX12, ALOX12B, and ALOX15, and type-2 diabetes mellitus (T2D), and found that ALOX12 and ALOX12B genetic variants increased susceptibility to T2D development, possibly though alterations in PUFA/ARA metabolism. 

Oxylipin Related Pathways. Prostaglandins, thromboxanes and leukotrienes are the classically described oxylipins involved in the pathogenesis of RA. The newer methods of LC/MS and NMR make it possible to identify several other oxylipins, e.g., 8-HETE, 12-HETE, and 12-HEPE are products of the 12-lypoxygenase pathway. Liagre et al. demonstrated the presence of 12-LOX in type B synoviocytes and found that IL-1β and TNF stimulation increased 12-HETE production, while IL-6 and IL-4 did not have the same effect [194]. This pathway was also studied by Kronke et al., who showed that the deletion of 12/15-LOX in two models of arthritis (the K/BxN serum-transfer and a TNF transgenic mouse model) led to uncontrolled inflammation and tissue damage [195]. LTB4 and 5-HETE are products of ARA via 5-LOX pathway; 5- and 15-LOX are expressed in both OA and RA synovium in the lining and sublining macrophages, neutrophils, and mast cells, and have been shown to be involved in RA pathogenesis, promoting inflammation [196]. 

Short Chain Fatty Acids (SCFA) are byproducts of the metabolism of dietary fiber by the gut microbiome. They modulate immune and inflammatory responses via the activation of free fatty acid (FFA) receptors type 2 and 3 (FFA2 and FFA3 receptors) and G protein-coupled receptor 109A (GPR109A), and via inhibition of histone deacetylases (HDACs). A metabolomic study performed on a CIA rat model found decreased levels of acetate, propionate, butyrate, and valerate in fecal samples of arthritic rats compared to controls [197]. The administration of butyrate inhibited collagen-induced arthritis via Treg/IL10/Th17 axis [198].

Bile acids (BA) seem to have anti-inflammatory properties. Primary BAs are synthesized in the liver and are liberated in the gastrointestinal tract to help with lipid digestion. Gut bacteria metabolize primary bile acids and can deconjugate them, synthesizing secondary BAs. BAs have been detected in the systemic circulation, where their concentrations vary with diet, and have been related to insulin resistance [199]. High concentrations of BA can actually kill intestinal bacteria to prevent colonization, but they also regulate the mucosal immune functions through several receptors. A recent study showed the role of BA in the maintenance of the homeostasis of the mucosal immune function in the gut, through the vitamin D receptor [126]. An in vitro study found that taurolithocholic acid suppressed the expression of genes involved in mediating pro-inflammatory effects, phagocytosis, interactions with pathogens and autophagy, as well as the recruitment of immune cells, such as NK cells, neutrophils and T cells [200]. BA exert their actions through both specific and nonspecific receptors. The activation of TGR5 receptor by endogenous BA suppressed the production of LPS induced inflammatory cytokines in macrophages, while no effect was seen in macrophages that lacked this receptor [201,202]. Of interest, a study described an anti-inflammatory role of taurochenodeoxycholic acid in RA FLS [203]. Most studies focused on the effects of the BA on the gut mucosal immunity, and hence, future studies are needed to elucidate the roles of circulating BAs in disease states. 

Tryptophan metabolism. Tryptophan is an essential amino acid that must be provided in the diet. It has been described that microbes-derived tryptophan metabolites can exert systemic and anti-inflammatory effects [124]. Moreover, tryptophan and its catabolic metabolites generated through the kynurenine pathway are involved in inflammation. Kynurenine has known anti-inflammatory effects that are toxic to T cells and induce cell death by apoptosis. Kynurenine is formed from tryptophan by the activity of indoleamine 2, 3-dioxygenase (IDO). The activation of IDO is actively involved in the resolution of arthritis in mice associated with an increase in kynurenine metabolites [204]. Kynurenine itself has been identified as a ligand for the aryl hydrocarbon receptor, which is important in the maturation of immune cells, and its addition promotes the differentiation of regulatory T cells and suppresses the differentiation of pathogenic Th17 cells [205].

## 4. Studies of Beneficial Effect of Diet in RA

In spite of the growing evidence of the relationship between diet and RA symptoms, research in the field is still limited to mostly observational studies; however, there are quite a few interventional studies in which diet has been evaluated as a strategy to improve RA symptoms. A detailed review of the studies can be found here [206,207,208]. Most interventions combine a diet with high intake of vegetables, fruit, and antioxidants with periods of fasting. The outcomes used in the majority of the studies include tender and swollen joint, disability index score, general health assessment scores, and a few inflammatory markers that can be quantified in blood, i.e., C reactive protein, erythrocyte sedimentation rate and proinflammatory cytokine (IL-6, TNF alpha).

These dietary intervention studies don’t shed any light on the specific metabolites that are responsible for the effect, or their mechanism of action. One study evaluated the anti-inflammatory effects of a low ARA diet and fish oil in patients with RA. Besides the usual outcomes, this study also quantified fatty acids and eicosanoids, using radioimmunoassay and gas chromatography coupled with MS [209]. They found that the diet improved RA clinical signs. In terms of fatty acid changes, they observed an enrichment of eicosapentaenoic acid in erythrocyte lipids and lower formation of urinary leukotriene B(4), 11-dehydro-thromboxane B(2) and prostaglandin metabolites in patients receiving the fish oil diet, especially when fish oil was given for a longer period of time (up to 8 months). 

The concentrations of plasma phospholipid related fatty acids were evaluated after a vegan and lacto-vegetarian diet intervention in RA patients. It was found that 20:3n-6 and 20:4n-6 were significantly reduced after 3.5 months of vegan diet, but the concentration increased to baseline values with a lactovegetarian diet. Also, 20:5n-3 was significantly reduced after both vegan and lactovegetarian diet periods. However, no significant difference in fatty acid concentrations was detected between diet responders and diet nonresponders after both diet periods, which suggests that the changes in the fatty acid concentrations were not in response to diet [210]. 

Another study in an RA Swedish population found that EPA and DHA were both increased in erythrocytes after a blue mussel diet compared to a control diet, resulting in a decrease of n-6 PUFA ARA (20:4 n-6) and dihomo-gamma-linolenic acid (DGLA; 20:3 n−6), as well as a small decrease in saturated fatty acids and the monounsaturated fatty acid palmitoleic acid (16:1 n-7). Baseline EPA and DHA levels in this population were higher than in healthy men and women, but this group had already reported that the RA population from the studied area had a higher fish and shellfish intake compared to the general population of Sweden [211]. In contrast to these findings, erythrocyte levels of α-linolenic acid (ALA; 18:3n3), EPA 20:5n3, and the omega-3 index (EPA plus DHA) were found to be significantly lower in RA patients compared to healthy controls in a Korean population, although diet was not accounted for in this study. In addition, EPA and ALA were negatively associated with the risk of RA in Korean women [212].

Additionally, the study of other autoimmune diseases does not help in establishing a link between diet and metabolites with their pro or anti-inflammatory effect. In Crohn’s disease, for instance, animal studies suggest the potential beneficial effect of short chain fatty acids, tryptophan, arginine, and glutamine due to their roles in the modulation of the immune system, but no clinical studies have been performed to date [213]. A cross-sectional metabolomics study also found decreased levels of essential PUFA in patients with lupus, but diet was not taken into account in this study [214]. Further studies are needed before we can make conclusions about the role of diet in the levels of circulating and local metabolites and their relationship with clinical outcomes in RA and other autoimmune diseases. 

## 5. Conclusions

Metabolomics studies have clearly shown that there is an alteration of the metabolic profile in patients with RA which may be related to the pathogenesis of the disease, but also to exposure to external factors, since the levels of the metabolites are influenced by several factors, including genetic ones, diet, sex, drugs, comorbidities, and microbiome (Figure 1). The possibility of altering some of these factors represents an attractive approach for future therapeutically interventions. Diet is a modifiable factor, and studies have shown that it can be effective in improving RA symptoms. Understanding the complex relation between diet, metabolites, microbiome, and disease status is still an ongoing process, but existing studies are promising. The field of RA needs more studies, including mendelian randomization studies and randomized clinical trials, combining the use of metabolomics, transcriptomics, and the microbiome to help understand how these elements interact with each other, identify patients who would benefit from a dietary intervention, and design the correct intervention.

## Figures and Tables

**Figure 1 cells-09-00827-f001:**
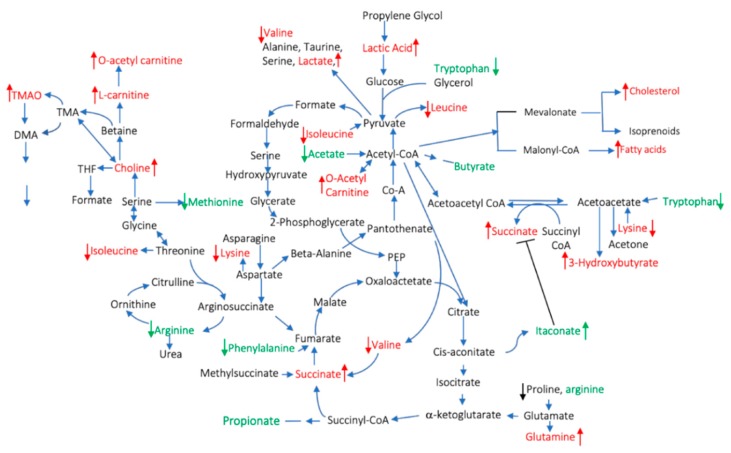
Pro- and anti-inflammatory circulating metabolites described in rheumatoid arthritis (RA) patients. The red color indicates pro- and the green indicates anti-inflammatory metabolites. The arrows indicate increases/decreased concentrations of the metabolites compared to healthy controls.

**Figure 2 cells-09-00827-f002:**
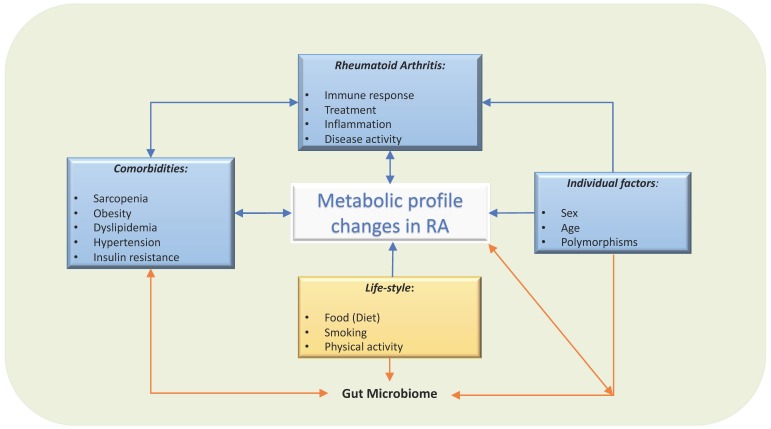
Factors involved in circulating metabolic profile in patients with RA. Several factors influence the circulating metabolites levels. Not only dietary factors or local synovial metabolites, but also comorbidities, treatment and individual factors, such as sex, age and genetics, will modify their metabolism, gut microbiome, and therefore, the circulating metabolic profile.

**Figure 3 cells-09-00827-f003:**
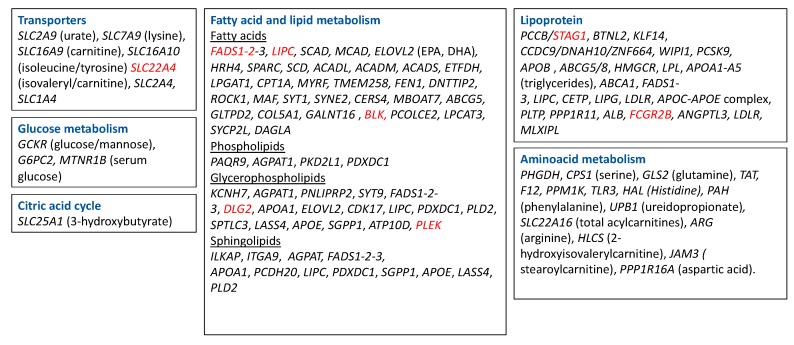
Metabolism-related genes described in GWAS. Highlighted in red are the genes that were found to be associated with RA. SLC2A9—Solute Carrier Family 2 Member 9; SLC7A9—Solute Carrier Family 7 Member 9; SLC16A9—Solute Carrier Family 16 Member 9; SLC16A10—Solute Carrier Family 16 Member 10; SLC22A4—Solute Carrier Family 22 Member 4; SLC22A4—Solute Carrier Family 2 Member 4; SCL1A4—Solute Carrier Family 1 Member 4; SLC25A1—Solute Carrier Family 25 Member 1; FADS—fatty acid desaturase; LIPC—Hepatic Triacylglycerol Lipase; SCAD—Short-chain acyl-CoA dehydrogenase; MCAD—Medium-chain acyl-CoA dehydrogenase; ELOVL2—Fatty Acid Elongase 2; HRH 4—Histamine Receptor 4; SPARC—Secreted Protein Acidic And Cysteine Rich; SPTLC3—Serine Palmitoyltransferase Long Chain Base Subunit 3; LASS4—Ceramide Synthase 4, SGPP1—Sphingosine-1-phosphate, ATP10D—ATPase Phospholipid Transporting 10D (Putative), SCD—Stearoyl-CoA Desaturase; ACADL—Acyl-CoA Dehydrogenase Long Chain; ACADM—Acyl-CoA Dehydrogenase Medium Chain; ACADS—Acyl-CoA Dehydrogenase Short Chain; ETFDH—Electron Transfer Flavoprotein Dehydrogenase; LPGAT1—Lysophosphatidylglycerol Acyltransferase 1; CPT1—Carnitine Palmitoyltransferase 1; PHGDH—Phosphoglycerate Dehydrogenase; CPS1—Carbamoyl-Phosphate Synthase 1; GLS2—glutaminase; EPA—eicosapentaenoic acid; DHA—docosahexaenoic acid; GCKR—Glucokinase Regulator; APOA—Apolipoprotein; MYRF—Myelin Regulatory Factor; TMEM258—transmembrane protein 258; FEN1—flap structure-specific endonuclease 1; DNTTIP2—deoxynucleotidyl transferase terminal interacting protein 2; ROCK1—Rho Associated Coiled-Coil Containing Protein Kinase 1; MAF—MAF BZIP Transcription Facto; SYT1—Synaptotagmin 1; SYNE2—Spectrin Repeat Containing Nuclear Envelope Protein 2; CERS4—Ceramide Synthase 4; MBOAT7—Membrane Bound O-Acyltransferase Domain Containing 7; ABCG5—ATP Binding Cassette Subfamily G Member 5; GLTPD2—Glycolipid Transfer Protein Domain Containing 2; COL5A1—Collagen Type V Alpha 1 Chain; GALNT16—Polypeptide N-Acetylgalactosaminyltransferase 16; BLK—BLK Proto-Oncogene, Src Family Tyrosine Kinase; PCOLCE2—Procollagen C-Endopeptidase Enhancer 2; LPCAT3—Lysophosphatidylcholine Acyltransferase 3; SYCP2L—Synaptonemal Complex Protein 2 Like; DAGLA—Diacylglycerol Lipase Alpha; PAQR9—Progestin And AdipoQ Receptor Family Member 9; AGPAT1—1-Acylglycerol-3-Phosphate O-Acyltransferase 1; PKD2L1—Polycystin 2 Like 1, Transient Receptor Potential Cation Channel, PDXDC1—yridoxal Dependent Decarboxylase Domain Containing 1; KCNH7—Potassium Voltage-Gated Channel Subfamily H Member 7; PNLIPRP2—Pancreatic Lipase Related Protein 2 (Gene/Pseudogene); SYT9—Synaptotagmin 9; DLG2—Discs Large MAGUK Scaffold Protein 2; CDK17—Cyclin Dependent Kinase 17; PDXDC1—Pyridoxal Dependent Decarboxylase Domain Containing 1; PLD2—Phospholipase D2; APOE—Apolipoprotein E; ILKAP—ILK Associated Serine/Threonine Phosphatase; ITGA9—Integrin Subunit Alpha 9; PCDH20—Protocadherin 20; GCKR—Glucokinase Regulator; G6PC2—Glucose-6-Phosphatase Catalytic Subunit 2; MTNR1B—Melatonin Receptor 1B; PCCB—Propionyl-CoA Carboxylase Subunit Beta; STAG1—Stromal Antigen 1; BTNL2—Butyrophilin Like 2, KLF14—Kruppel Like Factor 14, CCDC9—Coiled-Coil Domain Containing 9; DNAH10—Dynein Axonemal Heavy Chain 10; ZNF664—Zinc Finger Protein 664; WIPI1-WD Repeat Domain, Phosphoinositide Interacting 1; PCSK9 -, Proprotein Convertase Subtilisin/Kexin Type 9; APOB—Apolipoprotein B; HMGCR—3-Hydroxy-3-Methylglutaryl-CoA Reductase; LPL—Lipoprotein Lipase; ABCA1—ATP Binding Cassette Subfamily A Member 1; CETP—Cholesteryl Ester Transfer Protein; LIPG—Lipase G, Endothelial Type; LDLR—Low Density Lipoprotein Receptor; PLTP—Phospholipid Transfer Protein; PPP1R11—Protein Phosphatase 1 Regulatory Inhibitor Subunit 11; ALB—Albumin; FCGR2B—Fc Fragment Of IgG Receptor IIb; ANGPTL3—Angiopoietin Like 3; MLXIPL—MLX Interacting Protein Like; GLS2—Glutaminase 2, TAT—Tyrosine Aminotransferase; F12—; Coagulation Factor XII; PPM1K—Protein Phosphatase, Mg2+/Mn2+ Dependent 1K, TLR3—Toll Like Receptor 3; HAL—Histidine Ammonia-Lyase; PAH—Phenylalanine Hydroxylase; UPB1—Beta-Ureidopropionase 1; SLC22A16—Solute Carrier Family 22 Member 16; ARG—Arginase; HLCS—Holocarboxylase Synthetase; JAM3—Junctional Adhesion Molecule 3; PPP1R16A—Protein Phosphatase 1 Regulatory Subunit 16A.

**Figure 4 cells-09-00827-f004:**
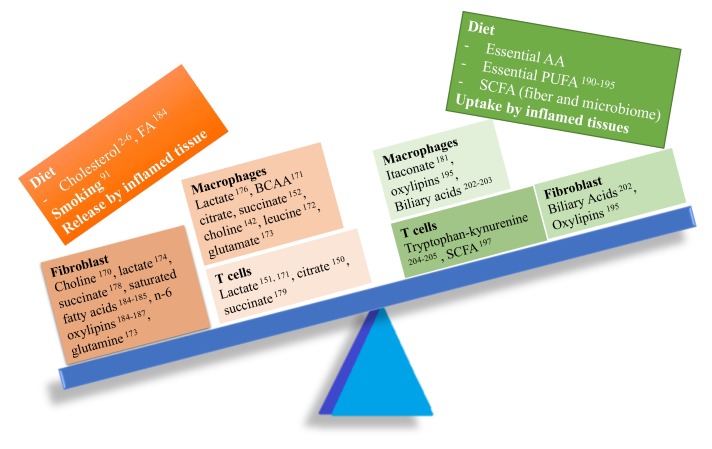
Imbalance between pro- and anti-inflammatory metabolites in RA. Several pro-inflammatory metabolites (left side of the balance) might play a key role in RA pathogenesis modulating the function of several cell types involved in synovial inflammation.

**Figure 5 cells-09-00827-f005:**
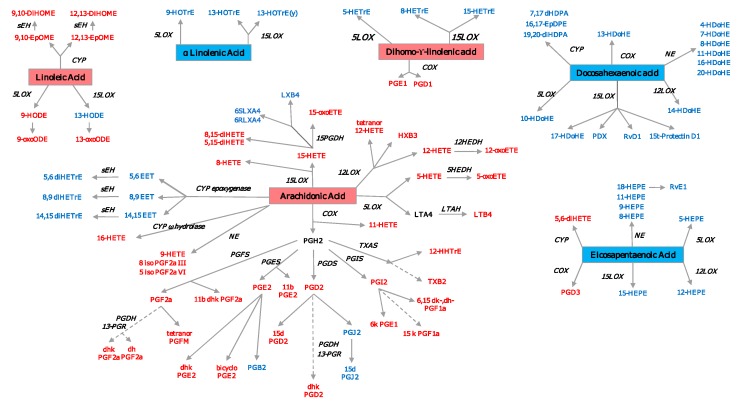
Oxylipin derived from PUFA. Pro-inflammatory oxylipins are marked in red, while anti-inflammatory ones are marked in blue. The precursor n3-PUFAs are marked in a red square, while the n6-PUFAs are marked in a blue square. COX—cyclooxygenase; LOX—lypooxigenase; CYP—cytochrome P450; NE—nonenzymatic; PGFS—prostaglandin F synthase; PGES—prostaglandin E synthase; PGDS—prostaglandin D synthase; PGIS—prostaglandin I synthase; TXAS—thromboxane A2 synthase; LTAH—leukotriene A4 hydrolase; MDB—membrane dipeptidase; HEDH—hydroxyeicosanoid dehydrogenase; PGDH—hydroxyprostaglandin dehydrogenase; 13-PGR—15-ketoprostaglandin∆13 reductase; sEH—soluble epoxide hydrolase. A list with all the oxylipins can be found in Appendix A.

**Table 1 cells-09-00827-t001:** Metabolic profile changes in plasma or serum of patients diagnosed with rheumatoid arthritis (RA). DMARD: disease-modifying antirheumatic drugs; GC: glucocorticoids; pSS primary Sjogren syndrome.

Type of Study	Number of Participants	Metabolite Changes
Plasma		
Prospective. RA patients vs. controls	47 RA patients on DMARDs (23 active and 24 in remission) and 51 controls. Sample collected at 0, 2, 4 weeks and 6, 12 months.	Elevated metabolites in RA patients compared to controls: choline, cholesterol, acetylated glycoprotein, lactate, and unsaturated lipid. Decreased HDL in RA patients compared to controls [2]
Cross-sectional	24 RA patients on methotrexate and less than 10 mg prednisolone daily	Positive correlation with fatigue in RA: Fructose, arachidonic acid (ARA), glycerol-3-phosphate, indole-3-acetic acid, and proline. Negative correlation with fatigue in RA: 2-oxoisocaproate, cystine, hydroxyproline, decosahexaenoic acid, tryptophan, pipecolic acid, valine, ornithine, arginine, urea, tyrosine, and linoleic acid [3]
Cross-sectional. RA patients vs. control	132 established RA patients and 104 controls	Metabolites increased in RA vs. control: prolyglycine. Metabolites decreased in RA vs. control: 4-methyl-2-oxopentanoate, 3-methyl-2-oxovalerate, and sarcosine. * Steroids in those with past corticosteroids treatment vs. those who never received them or are currently taking them [4]
Serum		
Cross-sectional. RA patients vs. controls	14 healthy controls 16 established RA patients, and two groups of early RA patients (89 and 127 RA patients)	High in RA patients compared to controls: 3-hydroxybutyrate, lactate, acetylglycine, taurine, glucose. Low in RA patients compared to healthy controls: LDL-CH3, LDL-CH2, alanine, methylguanidine, and lipid [5]
Cross-sectional. RA patients vs. controls	33 established RA patients and 32 controls	Metabolites increased in RA compared to controls: glycerol, citrate, pyruvate, cholesterol, fatty acids. Metabolites decreased in RA compared to controls: glucose, urate, alanine, serine, methionine, threonine, leucine, valine, isoleucine, aspartate, phenylalanine, tyrosine, proline, and urea [6]
Cross-sectional. RA on GC vs. RA that did not receive GC	281 RA patients 73 Males taking GC 42 Females taking GC	Higher in women on GC: lysophosphatidylcholines and lysophosphatidylethanolamines. In men, lysophospholipids levels were similar between GC users and nonusers [7]
Cross-sectional. RA and pSS patients vs. controls	30 active RA patients and 30 pSS as a disease control 32 controls	Metabolites increase in RA vs. pSS and control: L-Leucine, L-phenylalanine, glutamic acid, and L-proline, 4-methoxyphenylacetic acid. Metabolites decrease in RA vs. pSS and control: Tryptophan, argininosuccinic acid, and capric acid [8]

**Table 2 cells-09-00827-t002:** Food intake and candidate biomarkers identified by mass spectrometry (MS) and/or nuclear magnetic resonance (NMR).

Type of Food	Sample Type	Candidate Biomarker Metabolite
Meat (red meat, low-fat meat, chicken)	Urine Plasma	1-Methylhistidine; 3-methylhistidine; acetyl carnitine; creatinine; taurine; carnitine; trimethylamine N-oxide; creatine; histidine; urea; anserine; carnosine; guanidoacetate [19,22,23,24,25]
Beef	Plasma	β-Alanine; 4-hydroxyproline; 2-aminoadipic acid; leucine [26]
Fish	Urine Plasma	Trimethylamine N-oxide; anserine; 1-methylhistidine; 3-carboxy-4-methyl-5-propyl-2- furanpropanoic acid; docosahexaenoic acid (DHA); eicosapentaenoic acid (EPA); 1-docosahexaenoylglycero- phosphocholine; cetoleic acid [25,26,27,28,29,30,31]
Vegetables		
Vegetarian and lactovegetarian diet)	Urine	p-Hydroxyphenylacetate Hippurate; phenylacetylglutamine; lysine; hippurate; N-acetyl glycoprotein; succinate [19,32,33]
Broccoli	Urine	Ascorbate; tetronic acids; l-xylonate/l-lyxonate; naringenin glucuronide [28]
Onion	Urine	*N*-acetyl-S-(1Z)-propenyl-cysteine-sulfoxide 4-Ethyl-5-amino-pyrocatechol [34]
Lettuce, spinach, green peppers	Serum	3-Carboxy-4-methyl-5-propyl-2-furanpropanoic [30]
Cabbage, brussels sprouts, pointed cabbage	Urine	*N*-acetyl-*S*-(*N*-3-methylthiopropyl)cysteine; *N*-acetyl-*S*-(*N*-allylthiocarbamoyl)cysteine; iberin *N*-acetyl-cysteine; erucin *N*-acetyl-cysteine; *N*-acetyl-(*N*′ -benzylthiocarbamoyl)-cysteine; sulforaphane *N*-acetyl-cysteine; sulforaphane *N*-cysteine,3-Hydroxy-hippuric acid sulfate; 3-hydroxy-hippuric acid; iberin N-acetyl-cysteine [29]
Fruit		
Apples and pears	Urine	Phloretin [35,36]; rhamnitol [34]
Citrus	Urine	Proline betaine; limonene 8,9-diol glucuronide; nootkatone 13,14-diol glucuronide; hesperetin 3′-O-glucuronide; hydroxyproline betaine; *N*-methyltyramine sulfate; naringenin 7-O-glucuronide; stachydrine; scyllo- and chiro-inositol [28,30,35,36,37,38,39,40]
Orange juice	Urine	*N*-methyl proline; methyl glucopyranoside (α+β); stachydrine; betonicine; *N*-acetyl putrescine; dihydroferulic acid [41]
Raspberries	Urine	Sulfonated caffeic acid; methyl-epicatechin sulfate; 3-hydroxyhippuric acid; naringenin glucuronide; ascorbate [28]
Strawberries	Urine	4-Hydroxyhippuric acid; 4-hydroxy-2,5-dimethyl- 3(2H)-furanone (furaneol) glucuronide; pelargonin glucuronide; p-coumaric acid sulfate; dihydrokaempferol glucuronide; furaneol sulfate; 2,5-dimethyl-4-methoxy-2,3-dihydro-3-furanone (mesifurane); mesifurane sulfate; leucopelargonidin; catechin sulfate [28]
Cereals		
Whole-grain rye	Urine	Alkylresorcinol metabolites; caffeic acid sulfate; hydroxyhydroxyphenyl acetamide sulfate; 3,5-dihydroxyphenylpropionic acid sulfate; hydroxyphenyl acetamide sulfate [31]
Whole-grain sourdough rye bread	Urine Plasma	Benzoxazinoid derivatives; hydroxylated phenyl acetamide derivatives; sulfonated hydroxyl-*N*-(2-hydroxyphenyl) acetamide; *N*-(2-hydroxyphenyl)acetamide; 2,4-dihydroxy- 1,4-benzoxazin-3-one; 1,3-benzoxaxazol-2-one [42,43]
Whole-grain bread	Urine	Glucuronidated alk(en)ylresorcinols; 2-hydroxy-*N*-(2-hydroxyphenyl) acetamide; 2-hydroxy-1,4-benzoxazin-3-one glycoside; 3-(3,5-dihydroxyphenyl) propanoic acid glucuronide; 5-(3,5-dihydroxyphenyl) pentanoic acid sulfate; dihydroferulic acid sulfate; enterolactone glucuronide; pyrraline; 3-indolecarboxylic acid glucuronide; 2,8-dihydroxyquinoline glucuronide [43]
Dairy products		
Cheese	Urine	Indoxyl sulfate; xanthurenic acid; tyramine sulfate; 4-hydroxyphenylacetic acid; isovalerylglutamic acid; acylglycines; 3-phenyllactic acid [44]
Butter	Urine	3-Phenyllactic; alanine, proline; pyroglutamic acid; methyl palmitate (15 or 2); pentadecanoate (15:0); 10-undecenoate (11:1n–1) [30]
Milk	Urine Serum Plasma	Trimethyl-*N*-aminovalerate; uridine; hydroxysphingomyelin C14:1; diacylphosphatidylcholine C28:1; lactose; galactose; galactonate; allantoin; hippurate; galactitol; galactono-1,5-lactone [44,45,46]
Beverages		
Coffee	Urine	Caffeic; chlorogenic acid; Dihydrocaffeic acid-3-O-sulfate; feruloylglycine [35,47] Atractyligenin glucuronide; diketopiperazine cyclo(isoleucyl-prolyl); trigonelline; paraxanthine; 1-methylxanthine, 1-methyluric acid, 1,7-dimethyluric acid, 1,3- or 3,7-dimethyluric acid; 1,3,7-trimethyluric acid; 5-acetylamino-6-formylamino-3-methyluracil [48]
	Serum/Plasma	Trigonelline (*N*′-methylnicotinate); quinate; 1-methylxanthine; paraxanthine; *N*-2-furoyl-glycine; catechol sulfate [30] Pathways: xanthine metabolism; benzoate metabolism; steroid; fatty acid metabolism (acylcholine); endocannabinoid [49]
Black tea	Urine	Hippuric acid; 1,3-dihydroxyphenyl-2-O-sulfate gallic; 4-O-methylgallic acids [35,50]
Black/Green tea	Urine	Hippuric acid; 1,3-dihydroxyphenyl-2-O-sulfate; hydroxybenzoic glycine conjugate; vanilloylglycine; pyrogallol-2-O-sulfate [51,52,53]
Wine	Urine	Tartaric acid, microbial-derived phenolic metabolites (5-(dihydroxyphenyl)-γ-valerolactones and 4-hydroxyl-5-(phenyl)-valeric acids) [54]
	Plasma	Gallic acid and ethylgallate metabolites; resveratrol and resveratrol microbial metabolites; 2,4-dihydroxybenzoic acid; (epi)catechin; valerolactone metabolites [55]
Other		
Walnuts	Urine	10-Hydroxy-decene-4,6-diynoic acid sulfate; tridecadienoic/tridecynoic acid glucuronide; sulfate conjugates of urolithin A; 3-indolecarboxylic acid glucuronide; 5-Hydroxyindole-3-acetic acid [29,56]
Peanuts	Urine	4-Vinylphenol sulfate; tryptophan betaine [30]
Cocoa	Urine	Theobromine metabolism (AMMU; 3-methyluric acid; 7-methylxanthine; 3-methylxanthine; 3,7-dimethyluric acid; theobromine). Polyphenol microbial metabolites [methoxyhydroxyphenylvalerolactone; glucuronide and sulfate conjugates of 5-(3′,4′ -dihydroxyphenyl)- valerolactone] [57,58]
Chocolate	Urine	6-Amino-5-[*N*-methylformylamino]-1-methyluracil; theobromine; 7-methyluric acid [29]

**Table 3 cells-09-00827-t003:** Metabolic profile modifications by drugs used in the treatment of RA.

Samples	Decreased	Increased
Methotrexate [59]		
Plasma	Taurine, aspartate, alanine, hypoxanthine, cytosine, uric acid, uracil, lactic acid, S-adenosyl-L-homocysteine, 5-formyltetrahydrofolate, 5-methyltetrahydrofolate.	Tryptophan, threonine, histidine, methionine, glycine, carnitine, guanine, and adenosine.
Glucocorticoids		
Serum [7]	None reported	Lysophosphatidylethanolamines and lysophosphatidylcholines (Females).
Plasma [61]	Asymmetric dimethyl arginine, symmetric dimethyl arginine	None reported.
Anti-tumor necrosis factor (TNF)		
Serum [69]	3-hydroxyisobutyrate, lysine, acetoacetate, acetylphosphocholine, creatine sn-glycero-3-phosphocholine, histidine, and phenylalanine.	Leucine, acetate, betaine, and formate.
Serum [70]	3-hydroxybutyrate.	Isoleucine, leucine, valine, alanine, glutamine, tyrosine, and glucose.
Urine [71]	Eanolamine, p-hydroxyphenylpyruvic acid, and phosphocreatine.	Hippuric acid, citrate, and lactic acid (Infliximab). Choline, phenylacetic acid, urea, creatine, and methylamine (Etanercept). Histamine, glutamine, phenylacetic acid, xanthine, xanthurenic acid, and creatinine.

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
