# Peer review of "Circulating Pro- and Anti-Inflammatory Metabolites and Its Potential Role in Rheumatoid Arthritis Pathogenesis"

_cells, 2020, doi:10.3390/cells9040827_

Round 1

Reviewer 1 Report

Summary

The review summarizes the impacts of metabolites on RA pathogenesis from various angles – diet, drugs, comorbities, sex, age, genetics and gut microbiome. This is a very timely and particularly relevant article as the field begins to appreciate how metabolism can directly impact the immune functions in various different contexts from autoimmune diseases to brain function and embryonic development.

Overall impression

The article does a great job summarizing the various factors that can contribute to changes in metabolites in RA and how the changes in metabolites affects RA. The only criticism is that often facts are just stated with no context, mechanism or speculation. When a reader reads such a review, they not just want to know what studies are out there in the field but also how this fit in context and what the author’s take on those studies were. RA being an autoimmune disease, the review is sparse on immune mechanisms. My concerns have been discussed in detail below.

Point by point concerns:

The review is very narrowly focused on RA. For the review to provide contextual information to the reader, I would advise that the authors include a brief section on the differences between benefits of nutrition and metabolites, found between RA compared to other autoimmune disorders. Are the benefits discussed in this review from various foods and metabolites specific to RA or are they generally true for all autoimmune diseases? Which are more true across the board for all autoimmune diseases and which are more specifically true in RA? Such a section could be included at the end of the review.

Line 98-104, “Using a targeted metabolomic approach, Fu et al compared the effect of oral glucocorticoids (GC) on serum polar lipids and observed an increase in lysophosphatidylcholines and lysophosphatidylethanolamines in females but not in male patients with RA. This was a cross-sectional study that included patients that were on concomitant treatment with conventional synthetic disease modifying antirheumatic drugs (csDMARDs -Methotrexate, Hydroxychloroquine, Leflunomide), which could act as confounders, although they were included in the statistical analysis.”

The authors state facts without any rationale as to why the observed fact may be true, such as in these lines. Please follow with rationale - what are the possible explanations for an increase in polar lipids, lysophosphatidylcholines and lysophosphatidylethanolamines with GC treatment? How does this increase contribute to immune dysfunction resulting that contributes to RA?

Again, in line 105-110, what do the authors think or what does the literature say about the link between arginine metabolism and cardiovascular risk in RA patients

Again, what do they think is the link between decreased levels of several aminoacids as well as other metabolites, taurine, aspartate, alanine, lactic acid, adenosine and guanine – between the immune system? What is known in the literature or what do the authors propose is the link between these metabolites and immune mechanisms – how is this related to RA? Why/how does methotrexate reverse these levels?

Similarly, in the next paragraph – can the authors introduce immunological context instead of just stating observations? How do they think TNFi therapy affects these metabolites etc?

For points 2-5, if the literature/previous studies don’t provide exact answers/mechanisms, can the authors speculate? (especially  for the Drugs section)

Section 3.3 – Why do RA patients often present with metabolic syndrome, obesity sarcopenia? The authors need to include immunological context. For eg, studies have shown obesity associates with increase in certain immune sub-populations in the fat tissue etc.. The authors need to include immunological/mechanistic context and not just state the facts.

Section 3.5 – This section only discusses the effects of smoking and exercise on metabolites. It needs to be tied in context with RA and the immune system. For eg: “smoking induces metabolites x,y,z of which x has been shown to increase a in the immune system which has been related to increase susceptibility to RA.”

Section 3.6 – It only states that GWAS has shown links to metabolic pathways. The states that these metabolic pathways have shown links to RA. I think this section should discuss directly which GWAS loci have been directly implicated in RA and what pathways these loci have been show to impact in literature. For eg: “Loci x has been shown to link with RA patients in GWAS studies (Ref y, z, a..). This Loci x has been shown to directly impact proteins or pathways b, d, e etc.. (Ref f, g..). Pathways b,d, e have shown to result in higher metabolites which have been linked directly to RA.”. Or else this paragraph only states GWAS studies have turned up metabolic pathway loci (which is obvious and not much use to a readers who is hoping to learn specifically about RA pathogenesis via diet from your paper).

Line 286: No expansion for IRG1. It is difficult to understand what IRG1 stands for, reading that line. Please specify that it blocks BCAT1 activity. Only specifying that it is a leucine analog does not get the point across.

Paragraph for BCAA, Lines 279-291: It states that there are decreased levels of BCAA in RA patients. It then states that BCAA metabolism by BCAT1 results in inflammation and inhibit BCAT1 alleviates RA. – This is counterintuitive, why then do RA patients already have lower levels of BCAA?

In section 4.2, the following sub-headings are not bolded - Saturated and Polyunsaturated Fatty Acids Related Metabolites, Oxylipin Related Pathways, Short Chain Fatty Acids (SCFA)

The review is also missing an Immune system section. While I agree that there are reviews out there already about how diet & metabolites directly impact the immune system, this article should have atleast a brief paragraph about what metabolites have been linked to RA and what immune effects (eg what cell subset/cytokine/immune-pathway does it impact?) those metabolites have. This might be better done as a table as well. The paragraph could then refer the reader to more in-depth immune function-metabolite reviews out there.

Figure 1: The figure overemphasizes the role of gut microbiota. Several studies have shown that diet can have direct impacts on immune system function that is independent of the gut microbiome. Genetics can have an impact on gut microbiota, direct immune function and directly on dietary metabolism independent of gut microbiota. Same with sex, age and any of the factors listed in figure 1. Please modify figure 1 to accurately reflect all factors. While the gut microbiome is emerging as a clear unappreciated factor in all physiological functions. An unjustified overemphasis in literature on the role of the gut microbiome has been counterproductive to science.

How I would modify the figure: I would have an outer circle of immune system linking to all factors including the gut microbiome with double ended arrows when warranted: For eg genetics can influence the immune system but not vice versa. But the gut microbiome can influence immune function and vive-versa.

Author Response

We thank the reviewer for the supportive and constructive criticisms, which have helped to improve our manuscript.

Summary

The review summarizes the impacts of metabolites on RA pathogenesis from various angles – diet, drugs, comorbities, sex, age, genetics and gut microbiome. This is a very timely and particularly relevant article as the field begins to appreciate how metabolism can directly impact the immune functions in various different contexts from autoimmune diseases to brain function and embryonic development.

Overall impression

The article does a great job summarizing the various factors that can contribute to changes in metabolites in RA and how the changes in metabolites affects RA. The only criticism is that often facts are just stated with no context, mechanism or speculation. When a reader reads such a review, they not just want to know what studies are out there in the field but also how this fit in context and what the author’s take on those studies were. RA being an autoimmune disease, the review is sparse on immune mechanisms. My concerns have been discussed in detail below.

Point by point concerns

Reviewer: The review is very narrowly focused on RA. For the review to provide contextual information to the reader, I would advise that the authors include a brief section on the differences between benefits of nutrition and metabolites, found between RA compared to other autoimmune disorders. Are the benefits discussed in this review from various foods and metabolites specific to RA or are they generally true for all autoimmune diseases? Which are more true across the board for all autoimmune diseases and which are more specifically true in RA? Such a section could be included at the end of the review.

Response: We would like to emphasize that the metabolites that we review in section 3 have been proposed to be pro- or anti-inflammatory but there isn’t enough evidence to support a causal relation between the metabolites and it’s beneficial/inflammatory role in RA or other diseases. In addition, the interventional dietary studies that have shown beneficial effect of diet in RA or in other diseases have not included metabolomics as a tool to identify the potential metabolites that could be associated with those effects, except for a few that we mention in section 4. To avoid confusion on this, we have deleted ‘and diet’ from the title of section 3 and we have also added the following paragraph at the end of section 4:

The study of other autoimmune diseases does not help either in establishing a link between diet and metabolites with their pro or anti-inflammatory effect. In Chron’s disease, for instance animal studies suggest the potential beneficial effect of SCFA, tryptophan, arginine and glutamine due to their roles in the modulation of the immune system, but no clinical studies have been performed. Cross-sectional metabolomics studies also found decreased levels of essential PUFA in patients with lupus but diet was not taken into account in this study. Further studies are needed before we can conclude the role of diet in the levels of circulating and local metabolites and their relationship with clinical outcomes in RA and other autoimmune disease.

Reviewer: Line 98-104, “Using a targeted metabolomic approach, Fu et al compared the effect of oral glucocorticoids (GC) on serum polar lipids and observed an increase in lysophosphatidylcholines (lysoPC) and lysophosphatidylethanolamines (lysoPE) in females but not in male patients with RA. This was a cross-sectional study that included patients that were on concomitant treatment with conventional synthetic disease modifying antirheumatic drugs (csDMARDs -Methotrexate, Hydroxychloroquine, Leflunomide), which could act as confounders, although they were included in the statistical analysis.”

The authors state facts without any rationale as to why the observed fact may be true, such as in these lines. Please follow with rationale - what are the possible explanations for an increase in polar lipids, lysophosphatidylcholines and lysophosphatidylethanolamines with GC treatment? How does this increase contribute to immune dysfunction resulting that contributes to RA?

Again, in line 105-110, what do the authors think or what does the literature say about the link between arginine metabolism and cardiovascular risk in RA patients

Again, what do they think is the link between decreased levels of several aminoacids as well as other metabolites, taurine, aspartate, alanine, lactic acid, adenosine and guanine – between the immune system? What is known in the literature or what do the authors propose is the link between these metabolites and immune mechanisms – how is this related to RA? Why/how does methotrexate reverse these levels?

Similarly, in the next paragraph – can the authors introduce immunological context instead of just stating observations? How do they think TNFi therapy affects these metabolites etc?

For points 2-5, if the literature/previous studies don’t provide exact answers/mechanisms, can the authors speculate? (especially for the Drugs section)

Response: We agree with the reviewer that all these drugs are changing key metabolites involved in immune response and that the study of these changes might help to understand RA pathogenesis since therapeutic effects of these drugs could potentially be driven by metabolic changes. We have tried to include more context as requested by the reviewer (please see section 2.2) and/or give more references to the readers.

Reviewer: Section 3.3 – Why do RA patients often present with metabolic syndrome, obesity sarcopenia?  The authors need to include immunological context. For eg, studies have shown obesity associates with increase in certain immune sub-populations in the fat tissue etc.. The authors need to include immunological/mechanistic context and not just state the facts.

Response: The causes of the presence of metabolic syndrome, obesity and sarcopenia are still not completely known but the reader can find more information in the literature 2-6. Even though both fat tissue and muscle can be sources of metabolites, it is unknown how much they can contribute to the pool of circulating metabolites.  For example, studies measuring the metabolomics profile in visceral adipose tissue and serum from obese patients found low correlations between serum and adipose tissue metabolites. We have expanded that section accordingly (please see section 2.3)

Reviewer: Section 3.5 – This section only discusses the effects of smoking and exercise on metabolites. It needs to be tied in context with RA and the immune system. For eg: “smoking induces metabolites x,y,z of which x has been shown to increase a in the immune system which has been related to increase susceptibility to RA.”

Response: we have expanded the smoking section according the reviewer’s suggestions (please see section 2.5)

Reviewer: Section 3.6 – It only states that GWAS has shown links to metabolic pathways. The states that these metabolic pathways have shown links to RA. I think this section should discuss directly which GWAS loci have been directly implicated in RA and what pathways these loci have been show to impact in literature. For eg: “Loci x has been shown to link with RA patients in GWAS studies (Ref y, z, a..). This Loci x has been shown to directly impact proteins or pathways b, d, e etc.. (Ref f, g..). Pathways b,d, e have shown to result in higher metabolites which have been linked directly to RA.”. Or else this paragraph only states GWAS studies have turned up metabolic pathway loci (which is obvious and not much use to a readers who is hoping to learn specifically about RA pathogenesis via diet from your paper).

Response: As the reviewer pointed out, several genome wide association studies (GWAS) described genetic loci associated with metabolic pathways and some of them have been associated with RA. We have expanded that section and added the figure 3.

Reviewer: Line 286: No expansion for IRG1. It is difficult to understand what IRG1 stands for, reading that line. Please specify that it blocks BCAT1 activity. Only specifying that it is a leucine analog does not get the point across. Paragraph for BCAA, Lines 279-291: It states that there are decreased levels of BCAA in RA patients. It then states that BCAA metabolism by BCAT1 results in inflammation and inhibit BCAT1 alleviates RA. – This is counterintuitive, why then do RA patients already have lower levels of BCAA?

Response: we have modified that paragraph after reviewing the literature, since, as pointed out by the reviewer, the paragraph was somehow confused. In addition, the relationship between circulating and local synovial metabolites is not known because it hasn’t been studied in paired serum and synovial samples, but the decreased levels of circulating branch chained amino acids in RA patients could be due to their use as substrates for energy production by the inflamed tissue.

Reviewer: In section 4.2, the following sub-headings are not bolded - Saturated and Polyunsaturated Fatty Acids Related Metabolites, Oxylipin Related Pathways, Short Chain Fatty Acids (SCFA):

Response: done, thank you.

Reviewer: The review is also missing an Immune system section. While I agree that there are reviews out there already about how diet & metabolites directly impact the immune system, this article should have at least a brief paragraph about what metabolites have been linked to RA and what immune effects (eg what cell subset/cytokine/immune-pathway does it impact?) those metabolites have. This might be better done as a table as well. The paragraph could then refer the reader to more in-depth immune function-metabolite reviews out there.

Response: As the reviewer suggested we have expanded section 3 to include references to more in depth immunometabolism reviews, more details about what cell types are being modulated by pro and anti-inflammatory metabolites and included the figure 4 to summarize these changes.

Reviewer: Figure 1: The figure overemphasizes the role of gut microbiota. Several studies have shown that diet can have direct impacts on immune system function that is independent of the gut microbiome. Genetics can have an impact on gut microbiota, direct immune function and directly on dietary metabolism independent of gut microbiota. Same with sex, age and any of the factors listed in figure 1. Please modify figure 1 to accurately reflect all factors. While the gut microbiome is emerging as a clear unappreciated factor in all physiological functions. An unjustified overemphasis in literature on the role of the gut microbiome has been counterproductive to science. How I would modify the figure: I would have an outer circle of immune system linking to all factors including the gut microbiome with double ended arrows when warranted: For eg genetics can influence the immune system but not vice versa. But the gut microbiome can influence immune function and vive-versa.

Response: We have modified the figure to better illustrate the relationship between the different factors and circulating metabolites.

Reviewer 2 Report

This article is very interesting, but I have some comments.
1. Add a table that will show how the use of DMARDs and the biological therapy effect on metabolites in serum, plasma or urine.
2. What metabolic pathways are affected by polymorphisms? Add this information to the main text in section 3.6 Genetics. Can you add one Figure that will be present in these associations - for one metabolic pathway or for all? Authors should also add information on which SNPs or SNPs are most relevant to metabolites
3. Please add a number that will show the role of pro-inflammatory metabolites in RA as well as anti-inflammatory metabolites in RA

Author Response

We thank the reviewers for the supportive and constructive criticisms, which have helped to improve our manuscript.

This article is very interesting, but I have some comments.

Reviewer: Add a table that will show how the use of DMARDs and the biological therapy effect on metabolites in serum, plasma or urine.

Response: We have added the table 3 that summarizes the effect of drugs on metabolic profile.

Reviewer: What metabolic pathways are affected by polymorphisms? Add this information to the main text in section 3.6 Genetics. Can you add one Figure that will be present in these associations - for one metabolic pathway or for all? Authors should also add information on which SNPs or SNPs are most relevant to metabolites

Response: We have added a paragraph and figure 3, with the required information.

Reviewer: Please add a number that will show the role of pro-inflammatory metabolites in RA as well as anti-inflammatory metabolites in RA

Response: We have included a new figure (figure 4) showing the imbalance between pro and anti-inflammatory metabolites in RA

Reviewer 3 Report

This is a lengthy review on circulating metabolites and potential roles in pathogenesis of RA. In general this was a descriptive list of studies of various metabolites, I would have liked to see the authors attempt to tie studies together (perhaps by diagrams?) and to provide more critical analysis of the various studies. For example, do genomic loci that associate with metabolite levels also associate with RA? (Do this by look-ups of publicly-available data.) A section on this would add new information to the paper, and allow some causal inferences to possibly be made. 

For people not familiar with biochemistry, the lists of metabolites would be a little suffocating. Some metabolic pathway diagrams would be useful.

There could be a section at the end proposing the types of studies needed to elucidate possible causal roles for the various metabolites in RA. (eg RCT, Mendelian randomization studies.)

Other comments:  

Please also provide a comprehensive list of definition of abbreviations. Line 309, please expand on 'since epigenetic genetics have recently been described in RA.' Lines 329-30 - an elevation in RA serum does not demonstrate that FFA are pro-inflammatory.

Author Response

We thank the reviewers for the supportive and constructive criticisms, which have helped to improve our manuscript.

Reviewer: This is a lengthy review on circulating metabolites and potential roles in pathogenesis of RA. In general, this was a descriptive list of studies of various metabolites, I would have liked to see the authors attempt to tie studies together (perhaps by diagrams?) and to provide more critical analysis of the various studies. For example, do genomic loci that associate with metabolite levels also associate with RA? (Do this by look-ups of publicly-available data.) A section on this would add new information to the paper and allow some causal inferences to possibly be made.

Response: We have now included new graphics (Figures 1 to 5), another table (Table 3), and more information in the text, to try to give a more critical analysis of the different studies.

Reviewer: For people not familiar with biochemistry, the lists of metabolites would be a little suffocating. Some metabolic pathway diagrams would be useful.

Response: We have added two new figures (Figure 1 and Figure 5) that includes key metabolic pathways and key metabolites to help the reader.

Reviewer: There could be a section at the end proposing the types of studies needed to elucidate possible causal roles for the various metabolites in RA. (eg RCT, Mendelian randomization studies.)

Unfortunately, we are not familiar with clinical research, so we are not in a good position to propose and discuss pros and cons of the different types of studies to elucidate possible causal roles. We have added the mendelian randomization studies at the end of the manuscript.

Other comments: 

Reviewer: Please also provide a comprehensive list of definition of abbreviations.

Response: Done.

Reviewer: Line 309, please expand on 'since epigenetic genetics have recently been described in RA.'

Response: we have clarified that sentence

Reviewer: Lines 329-30 - an elevation in RA serum does not demonstrate that FFA are pro-inflammatory.

Response: We agree with the reviewer and we have modified that sentence.

Round 2

Reviewer 1 Report

The authors have done a thorough and thoughtful job addressing all reviewer concerns. Context and interpretation has been added to all the details provided which makes the review more interesting, informative and easy to read. All the figures have been remade and are very well done to reflect the principles discussed. I believe this review will be a good and timely addition to the field.

A few minor points:

1- Line 133-139: So what was the conclusion of the study? Did low dose GC ultimately lower cardiovascular risk in RA patients?

2- Line 141-147: The results of the study are stated. But the study had two treatment groups: Methotrexate only or Methotrexate+Chinese herb: Were the results seen for both treatment groups? Please clarify in text.

3- Line 154-155: It says earlier in the paragraph and in table 3 that Methotrexate decreases tryptophan levels but then this line states that the decrease in tryptophan is reversed by Methotrexate. So that would imply that Methotrexate actually increased tryptophan levels. Please clarify.

4- For Fig 4: please include the references that were used to make the figure in the figure legend.

5- Line 610: typo - Chron's

Author Response

We thank the reviewer for the suggestions which have improved our manuscript.

The authors have done a thorough and thoughtful job addressing all reviewer concerns. Context and interpretation has been added to all the details provided which makes the review more interesting, informative and easy to read. All the figures have been remade and are very well done to reflect the principles discussed. I believe this review will be a good and timely addition to the field.

A few minor points:

1- Line 133-139: So what was the conclusion of the study? Did low dose GC ultimately lower cardiovascular risk in RA patients?

We have clarified this in the manuscript.

            The effect of low dose GC (< 10mg/day) on arginine metabolism and cardiovascular risk in RA patients was also studied using a targeted metabolomics approach 61. This A study from Australia that included 36 RA patients, 18 of which were on GC (GC users) and 18 that were not receiving GC (non-GC-users), found that asymmetric dimethyl arginine (ADMA), which is associated to endothelial dysfunction, , mono methyl arginine (MMA), arginine, and citrulline concentrations were higher in non-GC users than in controls. In contrast, patients on chronic GC treatment had lower ADMA and SDMA (symmetric dimethyl arginine), suggesting an improved endothelial function and protective effect of long-term effect treatment with GC on endothelium of RA patients in terms of cardiovascular disease, by modulating arginine metabolism. 

2- Line 141-147: The results of the study are stated. But the study had two treatment groups: Methotrexate only or Methotrexate+Chinese herb: Were the results seen for both treatment groups? Please clarify in text.

We have clarified this in the manuscript.

3- Line 154-155: It says earlier in the paragraph and in table 3 that Methotrexate decreases tryptophan levels but then this line states that the decrease in tryptophan is reversed by Methotrexate. So that would imply that Methotrexate actually increased tryptophan levels. Please clarify.

Tryptophan is low in patients before treatment and Methotrexate increases it. We have made the corrections in the table.

4- For Fig 4: please include the references that were used to make the figure in the figure legend.

We have included the references in the figure, please see manuscript figure 4.

5- Line 610: typo - Chron's

We have corrected the typo, thank you.

The study of other autoimmune diseases does not help either in establishing a link between diet and metabolites with their pro or anti-inflammatory effect. In Crohn’s disease, for instance

Reviewer 3 Report

The authors have satisfactorily addressed my previous comments.

Author Response

We thank the reviewer for the suggestions, they helped improve our manuscript.